# UHPLC/MS-Based Untargeted Metabolomics Reveals Metabolic Characteristics of Clinical Strain of *Mycoplasma bovis*

**DOI:** 10.3390/microorganisms11102602

**Published:** 2023-10-21

**Authors:** Fei Yang, Mengmeng Yang, Duoduo Si, Jialin Sun, Fan Liu, Yanrong Qi, Shenghu He, Yanan Guo

**Affiliations:** 1College of Animal Science and Technology, Ningxia University, Yinchuan 750021, China; yangfeiweiwuxian@126.com (F.Y.); 15809605110@163.com (M.Y.); s956173936@163.com (D.S.); 18300795598@163.com (J.S.); liuf2785@163.com (F.L.); hlxxmcyz@126.com (Y.Q.); 2Institute of Animal Sciences, Ningxia Academy of Agricultural and Forestry Sciences, Yinchuan 750002, China

**Keywords:** *Mycoplasma bovis*, non-targeted metabolomics, metabolic pathway

## Abstract

*Mycoplasma bovis* is a global concern for the cattle industry owing to its high rates of infection and resulting morbidity, but its pathogenesis remains poorly understood. Metabolic pathways and characteristics of *M. bovis* clinical strain were elucidated by comparing the differential expression of metabolites between *M. bovis* clinical strain NX114 and *M. bovis* international reference strain PG45. Metabolites of *M. bovis* in the logarithmic stage were analyzed based on the non-targeted metabolomic technology of ultra-high performance liquid chromatography-mass spectrometry (UHPLC-MS). We found 596 metabolites with variable expression, of which, 190 had substantial differences. Differential metabolite analysis of *M. bovis* NX114 showed organic acids and their derivatives, nucleosides, and nucleotide analogs as important components. We found O-Phospho-L-serine (SEP) as a potential signature metabolite and indicator of pathogenicity. The difference in nucleic acid metabolites reflects the difference in growth phenotypes between both strains of *M. bovis*. According to KEGG enrichment analysis, the ABC transporter synthesis route had the most differential metabolites of the first 15 differential enrichment pathways. This study reflects the species-specific differences between two strains of *M. bovis* and further enriches our understanding of its metabolism, paving the way for further research into its pathogenesis.

## 1. Introduction

*Mycoplasma bovis* is a species of bacteria that is a key pathogen in cattle due to its high rates of morbidity in pneumonia and other infections [1]. The pathogen is extremely infectious and may quickly spread throughout a herd. *M. bovis* causes respiratory tract infections, mastitis, arthritis, otitis media, and infections in the womb after childbirth, leading to up to 80% of infection-related deaths in cattle [2]. Despite being identified in 1961, the pathogenesis of *M. bovis* remains unknown, particularly its molecular etiology. The prevalence of this pathogen is increasing owing to a lack of efficient vaccinations and medications for the prevention and treatment of diseases caused by its infection [3].

Qualitative and quantitative analysis of primary metabolites, signaling molecules, hormones, and secondary metabolites of small molecules in the entire life cycle or specific physiological cycles of microorganisms is an important field in systems biology [4]. The primary objective is to quantify the many dynamic reactions of metabolites in living organisms to external triggers, pathophysiology shifts, and gene mutations [5]. Microbial metabolomics is widely employed in biology and biomedicine, especially for microbial identification [6], metabolic pathway [7], antibiotic resistance [8], functional gene research [9], enzyme discovery [10], and synthetic biology [11]. Among them, non-targeted metabolomic analysis utilizing ultra-high-performance liquid chromatography-mass spectrometry (UHPLC-QqQ/MS) technology has grown in popularity in microbial metabolomic studies [12]. Moreover, the substantial effectiveness of this technique has been demonstrated by the identification of numerous detectable ionized using non-targeted metabolomics methods, followed by the use of multiple tools for metabolite annotation and structural characterization, providing a high-through microbial metabolomics method for biomarker discovery [13].

Members of Mycoplasma, a class of autonomous microorganisms with a minimal genome, can be used as model organisms for constructing metabolite networks in vitro. Currently, detailed metabolic network maps have been created for some model strains, such as *E. coli* and yeast [14]. Research scholars revealed differences among the metabolisms of *M. bovis* and *Mycoplasma gallinum* by comparing metabolomics and suggesting new gene annotations; nevertheless, certain primary or secondary metabolic processes remain unknown [15]. In this study, we utilized UHPLC-MS to investigate the differences in active metabolites between *M. bovis* clinical strain NX114 and *M. bovis* international reference strain PG45, as well as to investigate the complex and diverse metabolic pathways and features of *M. bovis*. Analysis of multiple response changes with a broader dynamic range and greater accuracy were utilized to identify and discriminate clinically crucial strains of *M. bovis*, providing high-throughput and big data regarding *M. bovis* pathogenicity. We believe that this paper will contribute to the understanding of *M. bovis* biology and its implications for bovine health. Moreover, our findings will stimulate further research in this area and pave the way for novel therapeutic strategies.

## 2. Materials and Methods

### 2.1. Bacterial Strains and Media

*M. bovis* clinical strain NX114 (GenBank accession no. CP135997) was isolated from the lung of a diseased Holstein calf with pneumonia in Ningxia Province, China. *M. bovis* type strain PG45 (ATCC 25523) is the international reference strain of *M. bovis*, and it was donated by Professor Aizhen Guo of Huazhong Agricultural University of China. Both strains were stored in PPLO broth (BD Difco^TM^, San Jose, CA, USA) [16] with 40% glycerol (*v*/*v*) at −80 °C. Before utilization, both strains were cultured in PPLO broth.

### 2.2. Transmission EM

Samples were serially diluted tenfold at a ratio of 1:10 in DPBS (without Ca^2+^/Mg^2+^) before undergoing negative staining with 1% phosphotungstic acid. A grid with a carbon-coated formvar supporting film was added to for 5 min using a 15-L aliquot. Filter paper was used to absorb any surplus solution before the samples were dyed for 3 min and dried by air. Samples were viewed using an 80-kV electron microscope (JEM-1400 FLASH, Tokyo, Japan). Photographs were captured using a digital camera (EMSIS, Münster, Germany).

### 2.3. Determination of Growth Curve 

We inoculated the isolates of *M. bovis* NX114 and *M. bovis* strain PG45 were added into PPLO liquid medium (10% *w*/*v*). Then, the number of colony-forming units (CFU) were measured once per 0 h up to 72 h (*n* = 6). Finally, we plotted the *M. bovis* growth curve using the log CFU/mL parameter as an ordinate and the culture time of the bacterial solution as the abscissa.

### 2.4. Extraction of Metabolites from Strains

A vortex was used to fully combine a 100 µL samples of both strains *M. bovis* in the logarithmic stage with 400 µL of cool, 1:1 methanol acetonitrile. Next, the combination was processed using cold baths and sonication. The combination was then centrifuged at a speed of 14,000× *g* for 20 min at 4 °C after being incubated at −20 °C for 1 h. Following vacuum drying, the supernatants were analyzed using UHPLC-MS.

### 2.5. UHPLC-MS/MS Analysis

The sample was automatically injected into an injector at 4 °C throughout the analysis. Samples were analyzed with the SHIMADZU-LC30 instrument and ACQUITY UPLC^®^ HSS T3 column from Waters (Milford, MA, USA). The mobile phase included A: 0.1% FA in water and B: 100% acetonitrile (ACN) at a flow rate of 0.3 mL/min. The gradient was 0% buffer B for 2 min, then 48% in 4 min, 100% in 4 min, sustained for 2 min, and finally 0% for 0.1 min, with a 3-min re-equilibration interval. Electrospray ionization (ESI) with positive and negative modes was used independently for MS data gathering. The HESI source conditions were Spray Voltage: 3.8 kV (positive) and 3.2 kV (negative); Capillary Temperature: 320 °C; Sheath Gas (nitrogen) flow rate: 30 arb (arbitrary units); Aux Gas flow rate: 5 arb; Probe Heater Temp: 350 °C; S-Lens RF Level: 50. For complete MS, the device was configured to acquire across the *m*/*z* range 70–1050 Da. The complete MS scans were taken at a resolution of 70,000 at *m*/*z* 200, and the MS/MS scan at 17,500 at *m*/*z* 200. The maximal injection times for MS and MS/MS were established at 100 ms and 50 ms, respectively. Finally, MS2 had a 2 *m*/*z* isolation window, and a normalized collision energy (stepped) for fragmentation was set to 20, 30, and 40.

### 2.6. Data Preprocessing 

Data from the raw MS were aligned, corrected for retention time, and extracted using MS-DIAL (ver. 2.62) software. Metabolites were detected using precision mass (mass tolerance < 10 ppm) and MS/MS data (mass tolerance < 0.02 Da), which were compared to HMDB, massbank, and other public databases, alongside our self-built metabolite standard library. Only variables with at least 50% nonzero measurement values were retained in the extracted-ion features.

### 2.7. Multivariate Data Processing and Data Analysis

For all multivariate data analytics and modeling, R (version: 4.0.3) and R packages were utilized. Using Pareto scaling, the data were mean-centered. Principal component analysis (PCA), symmetrical partial least-square discriminant analysis (PLS-DA), and partial least-square discriminant analysis (OPLS-DA) were used to create models. With the use of permutation testing, all the evaluated models were checked for overfitting. R2X (cumulative) (perfect model: R2X (cum) = 1) and R2Y (cumulative) (perfect model: R2Y (cum) = 1) values were used to assess the models’ descriptive performance, while Q2 (cumulative) (perfect model: Q2 (cum) = 1) and a permutation test (*n* = 200) were used to assess their prediction performance. The permuted model should be unable to forecast classes; its R2 and Q2 values at the *Y*-axis intercept should be lower than the non-permuted model for Q2 and R2. OPLS-DA enabled the identification of distinguishing metabolites by utilizing the variable importance on projection (VIP). The VIP score value represents a variable’s contribution to sample discrimination across all classes. These scores are mathematically derived for each variable as a weighted sum of squares of PLS weights. The average VIP value is 1, and VIP values >1 are typically regarded as noteworthy. A high score corresponds to a good discrimination capacity and therefore serves as a criterion for biomarker selection.

The variable effect on projection (VIP) values from the OPLS-DA model and the two-tailed Student’s *t* test (*p* value) on the normalized raw data at the univariate analysis level were used to determine the discriminating metabolites. One-way analysis of variance (ANOVA) for multiple groups analysis was used to obtain the *p* value. Statistically-significant metabolites were those with VIP values larger than 1.0 and a *p* value lower than 0.05. The average mass response (area) ratio between two arbitrary classes was used to compute fold change. Cluster analysis, using the R package, was carried out using the detected differential metabolites. We used KEGG pathway analysis to identify the disturbed biological processes based on differential metabolite data (http://www.kegg.jp, accessed on 20 December 2022). Fisher’s exact test and FDR correction for multiple testing were used in KEGG enrichment studies. At the *p* < 0.05 level, enriched KEGG pathways were nominally statistically significant.

## 3. Results

### 3.1. Attributes of M. bovis

*M. bovis* may have a spherical, oval, or irregular polygonal shape, as transmission electron microscopy has shown. Individuals from *M. bovis* NX114 (Figure 1A,B) were slightly larger than those of PG45 (Figure 1C,D). Complete and clear cell membranes were observed on the surfaces of both strains of *M. bovis*, and no cell walls were found.

### 3.2. Measurement of Growth Curve

The results demonstrate that *M. bovis* NX114 and PG45 have comparable growth patterns; there was little variance in growth rate or proliferation ability across both strains. Both reached logarithmic phase after 36 h of culture (Figure 2). 

### 3.3. Screening of Differential Metabolites of M. bovis

The UHPLC-Q-Exactive LC-MS approach was used in this work to characterize the metabolic characteristics of *M. bovis* in its log phase using non-targeted metabolomics. In *M. bovis* samples cultivated to log phase, 596 divergent metabolites were identified based on positive and negative ion collection techniques, of which 190 were significantly different compounds (Appendix A). Quality control (QC) sample two-dimensional principal component analysis (2D PCA) score plots revealed an overall separation tendency between the *M. bovis* NX114 and PG45 strains (Figure 3A,B). The PCA model for *M. bovis* showed a trend towards complete separation, indicating a significant difference in the metabolic profile of *M. bovis* between the two strains. To accomplish visual clustering, we employed the orthogonal partial least squares discriminant analysis (OPLS-DA) model. The OPLS-DA model revealed that *M. bovis* NX114 and PG45 strains had a general separation (Figure 4A,B). OPLS-DA showed that differences between different strains of *M. bovis* could be reflected at the metabolic level. The permutation tests verified the reliability of these models (Figure 4C,D).

The volcano plot (Figure 4E,F) shows changing trends of differential metabolite expression in the *M. bovis* NX114 and PG45 strains. Filter criteria were FC > 1.5, FC < 0.667, and *p* value < 0.05; the vertically dotted line analysis marked log_2_ (1/1.5) and log_2_ (1.5). Each point reflects a specific metabolite, and its size corresponds to the VIP value. The VIP value increases as the dot gets bigger. Blue dots signify downregulated metabolites, and red dots signify upregulated metabolites. Utilizing univariate statistical analysis, different metabolites were examined.

### 3.4. Analysis of Differential Metabolite Expression of M. bovis

Significant variations in metabolite screening criteria were discovered using OPLS-DA with VIP > 1 and *p* < 0.05 (Student’s *t*-test). We utilized the expression of qualitatively analyzed significant difference metabolites for hierarchical clustering of *M. bovis* NX114 and PG45 (Figure 5A), where 53 were upregulated and 69 were downregulated in positive ion mode; 20 were upregulated, and 48 were downregulated, when the ions were negative (Appendix A). To investigate the connection between the differential metabolism of *M. bovis* NX114 and PG45, additional investigation of differential metabolites was carried out. The link between various metabolites was investigated using Pearson’s correlation analysis (Appendix A). A correlation coefficient matrix heat map was made using the top 10 metabolites with VIP levels (Figure 5B). Our findings demonstrate a close association between metabolites in similar or identical metabolic pathways. A Circos plot (Figure 5C) was used to visualize the data relationship, which was used to show the association between multiple metabolites.

### 3.5. Function Analysis of Differential Metabolites of M. bovis

A total of 190 different metabolites were classified and counted according to their structures and functions. The HMDB database revealed 109 significantly separate metabolites out of these. Several different metabolites were separated into 12 teams, including 23 organoheterocyclic compounds (21.1%) and 20 lipids and lipid-like molecules (18.35%). There were 18 benzenoids (16.51%); 12 organic acids and derivatives (11.01%); 11 phenylpropanoids and polyketides (10.99%); 7 organic oxygen compounds (6.42%); 6 nucleosides, nucleotides analogues, and organic nitrogen compounds (5.5%); 3 alkaloids andderivatives (2.75%); and 1 hydrocarbon, lignans neolignans and related compounds, and organosulfur compounds (0.92%) (Figure 6A).

Using Fisher’s exact test analysis and calculation of the significance amount of metabolite enrichment via each pathway, 85 pathways in the clinical strain of *M. bovis* NX114 were changed compared to those of the *M. bovis* international reference strain PG45 (Appendix A). KEGG ID mapping of metabolites with significant differences was performed to clarify functions of the metabolites and the relationships between metabolites. The top 30 most significant metabolic pathways were selected according to their *p* values and are presented as a bubble map (Figure 6B). The first 30 metabolic pathways mainly involved ABC transporters, neuroactive ligand–receptor interactions, ferroptosis, pyrimidine metabolism, arginine and proline metabolism, and urine metabolism.

To systemically study metabolic changes, the overall trend of metabolic pathways was analyzed using differential metabolite abundance. Differential abundance scores (DA scores) captured the trend of overall metabolite increase/decrease in the pathway relative to the *M. bovis* international reference strain PG45. The DA score was calculated for the differential metabolite annotation data, and the top 30 pathways were chosen for drawing pictures (Figure 6C). To investigate the functional relationships of the latent differential metabolite enrichment pathways, the functional interactions of these pathways were evaluated using Cytoscape (Figure 6D). The results showed that ABC transporters, neuroactive ligand–receptor interactions, and cofactor biosynthesis pathways are the hubs of other related pathways.

We used an enriched string diagram (Figure 6E) and a Sankey energy shunt diagram (Figure 6F) to visualize data flow for differential metabolites and KEGG pathway levels. The results indicated that for the 21 differential metabolites engaged in the top 15 differentially enriched pathways (Appendix A), the average expression of the differential metabolites in both strains is as shown in a box plot (Figure 7). Differential metabolites were observed between the two strains, with upregulated metabolites including O-Phospho-L-Serine (SEP) and melatonin (ML) (*p* < 0.05), and downregulated metabolites including uridine 5’-diphosphate (UDP), L-glutamic acid (Glu), glutathione (GSH), CMP, uridine 5’-monophosphate (UMP), riboflavin (B2), spermidine (SP), guanosine (Gsn), 2-isopropylmalic acid (2-IPMA) (*p* < 0.05). Additionally, N-acetylputrescine (NAP), ade-nine (Ade), alpha-tocopherol (at), flavin adenine dinucleotide (FAD) (*p* < 0.01), adenosine (Adn), uridine (Urd), trans-4-hydroxy-L-proline (T-4-Hyp), ophthalmate (OA) (*p* < 0.001), L-leucine (Leu), and guanine (G) did not show significant differences (*p* > 0.05).

## 4. Discussion

*Mycoplasma bovis* is a pathogen that is not well understood, but can be extremely harmful to cattle. Previous studies have demonstrated that *M. bovis* is capable of inducing host lesions through the production of secondary metabolites [17]. Metabolites secreted by *M. bovis* are important in the induction of immune responses in host cells [18,19]. Metabolites are intermediates or end products of microbial metabolic processes, which are closely related to the phenotypes of biological systems and control the regulation of phenotypic functions, so they better reflect the body phenotype than other “omics.” Moreover, sampling in the logarithmic phase for detection can obtain more metabolites, which are more representative to reflect the growth state of microorganisms [20,21]. In this study, we investigated the metabolic pathways and characteristics of *M. bovis* clinical strain NX114 in comparison to those of the international standard strain PG45 during the logarithmic growth phase.

### 4.1. Important Components of Metabolites of M. bovis Clinical Strain

The study utilized UHPLC-MS non-targeted metabolomic technology combined with multivariate statistical analysis to identify and analyze differential metabolites in the two strains. In recent years, researchers have used large-scale metabolomics analysis to accurately map the metabolic network of *Mycoplasma pneumonia* [22] and qualitative images of metabolic pathways [23], and they have constructed a genomic metabolic model based on *M. hyopneumoniae* genome sequencing [24]. However, there is not yet an analysis of potential metabolite differences between the two pathogenic *M. bovis* strains isolated in the same host range. Our findings revealed a total of 596 differentially expressed metabolites between the two strains, with 190 of them showing significant differences. These metabolites were classified and counted according to structure and function, and 109 differential metabolites were searched through the HMDB database and further divided into 12 categories.

The differences in metabolite types and contents among the top five organoheterocyclic compounds (lipids, lipid-like molecules, benzenoids, organic acids, and their derivatives, phenylpropanoids, and polyketides) were characterized in detail for the two strains of *M. bovis* (Figure 6A). The interactions between these diverse metabolites were discovered using Pearson’s correlation analysis to evaluate the correlations between them (Appendix A), which would assist in disclosing possible metabolic control. By constructing an association network of the differentially expressed metabolites (Figure 5C), we found that organoheterocyclic compounds, nucleosides, and nucleotide analogs, although not the highest in terms of the number of differential metabolites, most closely correlated with other classes. It was revealed that organoheterocyclic compounds, nucleosides, and nucleotide analogues are important components of metabolites of *M. bovis* clinical strain NX114, but there are still 81 differential metabolites that have not yet been characterized, and further identification of differentially expressed metabolites of unknown classification should be conducted qualitatively and quantitatively.

### 4.2. Trend of Differential Metabolites and Pathways at All Levels of M. bovis

The results of this study showed that 21 distinct metabolites were implicated in the first 15 significantly enriched pathways (Appendix A). The upregulated metabolites SEP and ML (*p* < 0.05) were mainly associated with aminoacyl-tRNA biosynthesis, cysteine, and methionine metabolism, circadian entrainment, and neuroactive ligand–receptor inter-actions. Downregulated UDP, Glu, GSH, CMP, UMP, B2, SP, Gsn, 2-IPMA (*p* < 0.05), NAP, Ade, at, FAD (*p* < 0.01), Adn, Urd, T-4-Hyp, OA (*p* < 0.001), and Leu and G (*p* > 0.05) were mainly associated with ABC transporters, aminoacyl-tRNA biosynthesis, arginine and proline metabolism, biosynthesis of cofactors, circadian entrainment, cysteine and methionine metabolism, ferroptosis, glutathione metabolism, neuroactive ligand–receptor interaction, purine metabolism, pyrimidine metabolism, riboflavin metabolism, valine, leucine, and isoleucine biosynthesis, vitamin digestion, and absorption are associated. Among these, enrichment of the ABC transporter synthesis pathway was the most significant and of the highest degree.

### 4.3. Potential Signature Metabolite and Possible Mechanisms of M. bovis

O-Phospho-L-serine (SEP), a molecule similar to the head group of phosphatidylserine, acts as a substrate for cysteine synthase (CysM) in vivo [25]. In this study, SEP was enriched in cysteine and methionine metabolism, trNA-aminoacyl biosynthesis pathways, and correlated with L-serine biosynthesis. However, SEP acts as a direct precursor of L-serine in the process of glycolysis pathway synthesis [26], whereas serine can only be phosphorylated when it is a protein component [27]. The significant difference in SEP abundance between the two strains of *M. bovis* in the present study suggests that it may be due to different phosphoserine phosphatase activities when phosphoserine is finally converted to L-serine. In addition, SEP (Log_2_FC = 3.0292) was significantly higher in *M. bovis* NX114 than in *M. bovis* PG45 (*p* = 0.017). In addition, previous studies have shown that the relative abundance of SEP in *M. bovis* PG45 strain is higher than that in Mycoplasma gallinarum AP3AS strain [15], but there is no significant difference between it and its transposon mutant strain [28]. In addition, studies have shown that an increase in SEP is related to a higher severity of some diseases and deaths [29,30]. Therefore, SEP has potential as a biomarker with high expression in *M. bovis*. Whether SEP can be used to distinguish between Mycoplasma of different species and *M. bovis* of different genotypes. Whether the accumulation of SEP can be used as an indicator to judge the severity of the disease after *M. bovis* infection should be further evaluated by metabolite quantitative test to improve understanding of the pathogenesis of *M. bovis*.

Gamma-glutamyl-cysteine-glycine (GSH), a low-molecular-weight mercaptan produced from glutamic acid, cysteine, and glycine, is one of these compounds [31]. Our results revealed that *M. bovis* NX114 had considerably reduced levels of GSH (*p* < 0.05) and Glu (*p* < 0.05), as well as an enrichment in the glutathione metabolic pathway and involvement in several growth factor transport and amino acid metabolism pathways. These metabolites may reflect potential species-specific differences between the two strains of *M. bovis* within the same host range. In addition, KEGG enrichment analysis indicated that low levels of GSH, Glu, and Alpha-tocopherol may be related to Ferroptosis. GSH is the core component of amino acid metabolism during iron death, and ferroptosis is an iron-dependent programmed cell death mode. It is distinguished by GSH depletion, lipid peroxide (ROS) buildup, and glutathione peroxidase 4 (GPX4) inactivation [32]. These results suggest that *M. bovis* NX114 induces Ferroptosis in host cells through an abnormal amino acid metabolism pathway.

### 4.4. Nucleic Acid Metabolite Differences Reflect Distinct Growth Characteristics of M. bovis

Mycoplasma lacks the enzymatic metabolic pathways required for purine synthesis and the whey acid metabolic pathway required for pyrimidine synthesis. Therefore, the need for exogenous nucleic acids and nucleic acid precursors of mycoplasma is an important nutritional metabolic feature that distinguishes mycoplasma from other micro-organisms and accurately reflects its growth state [33]. Previous studies have used metabolomics technology to identify significant differences in the major nucleotide bases, nucleic acids, and nucleotides of two kinds of mycoplasma infecting different hosts (poultry and cattle), which confirmed the different nutritional metabolism characteristics of mycoplasma [15]. In addition, an enzyme converting nucleotide monophosphate to nucleoside was annotated in *M. bovis*, and an enzyme with similar function was found in *M. gallisepticum* [34], but there was no such enzyme or an enzyme with similar function in *M. pneumoniae* [23]. In this study, there were differences in purine and pyrimidine metabolism between the two strains of *M. bovis*, and the nucleic acid metabolites accurately reflected the different growth states of the two strains of *M. bovis*, suggesting that there may be different functions and activity levels of nucleosidases in *M. bovis* and the differences in nucleic acid metabolism pathways may be the reason for the differences in growth phenotypes of the two strains.

### 4.5. Significant Involvement of Transport System in M. bovis Metabolism

Mycoplasmas are assumed to rely on their host for numerous nutrients; hence, a robust transport mechanism is required to deliver nutrients [35]. Among them, the ABC transport system serves as the primary means of substance exchange between mycoplasma and the external environment [36]. Metabolite and bioinformatics analyses revealed that *M. bovis* can take up and replenish deficient nucleotides and nucleotide precursors into cells via nucleotide ABC transporters [28]. Moreover, it has been suggested that the substrate binding the ATP-dependent ABC transport system involved in nucleoside uptake may be one of the compensatory transporters [28]. The ABC transporter synthesis pathway was enriched to the highest and most significant degree in the present study, which further demonstrates that ABC transporters play an important role in the metabolism of *M. bovis*. However, the statistical analysis revealed that the metabolites G (*p* < 0.05), as well as Urd and Ade (*p* < 0.001), were significantly downregulated. Based on the findings, *M. bovis* NX114 potentially exhibits a deficiency or dysfunction in the nucleotide ABC transporter, alongside a lack of compensatory pathways.

Furthermore, oligophosphoric and amino acid uptake is considered particularly important for the growth of *M. bovis*, as their preferred nutrient sources have been proposed to be amino acids or organic acids [37]. Therefore, the abundance of Leu, Glu, Gsn, and T-4-Hyp in *M. bovis* NX114 was low in this study, reflecting the life and growth of two strains through the absorption of nutrients via the ABC transport system.

## 5. Conclusions

In conclusion, this study revealed the differential expression of metabolites between the *M. bovis* clinical strain NX114 and *M. bovis* international reference strain PG45 during logarithmic growth. Based on UHPLC-MS non-targeted metabolomics techniques and multivariate statistical analysis, 596 differential metabolites were identified in this study, 190 of which were significant differential metabolites. These metabolites reflect the metabolic expression characteristics of *M. bovis* NX114, and the metabolic pathways involved, such as amino acid metabolism, nucleic acid metabolism, and transport system, reflect the different growth characteristics and potential species-specific differences of two strains of *M. bovis* with the same host range. These findings will stimulate further research in this area and pave the way for novel therapeutic strategies.

## Figures and Tables

**Figure 1 microorganisms-11-02602-f001:**
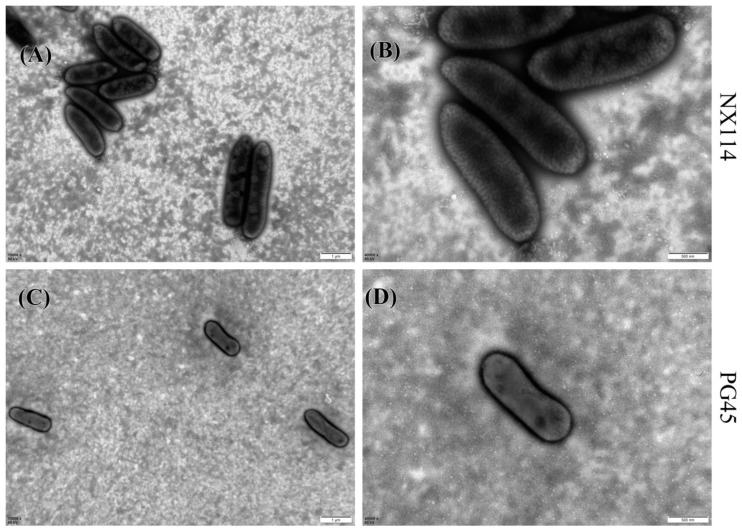
Morphology of *M. bovis* under transmission electron microscope after negative staining; (**A**,**C**) 15,000×; (**B**,**D**) 40,000×.

**Figure 2 microorganisms-11-02602-f002:**
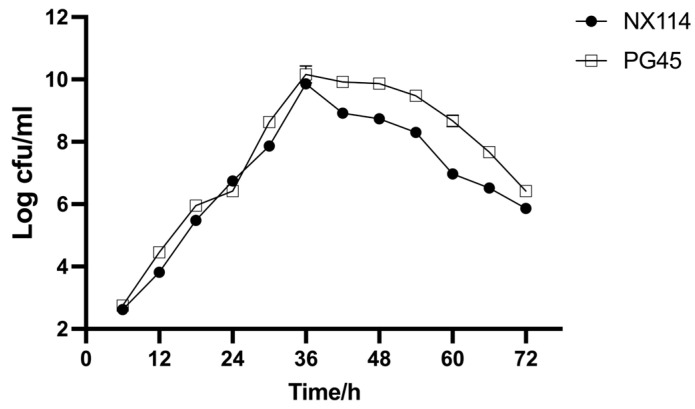
Growth curve of both strains of *M. bovis* in PPLO medium.

**Figure 3 microorganisms-11-02602-f003:**
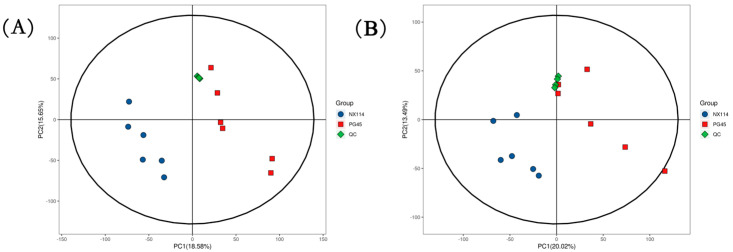
PCA score of *M. bovis* metabolomics analysis. (**A**,**B**) PCA score plots utilizing the metabolic profiles of both strains in positive and negative modes. ESI+: R2 = 0.586, ESI−: R2 = 0.569.

**Figure 4 microorganisms-11-02602-f004:**
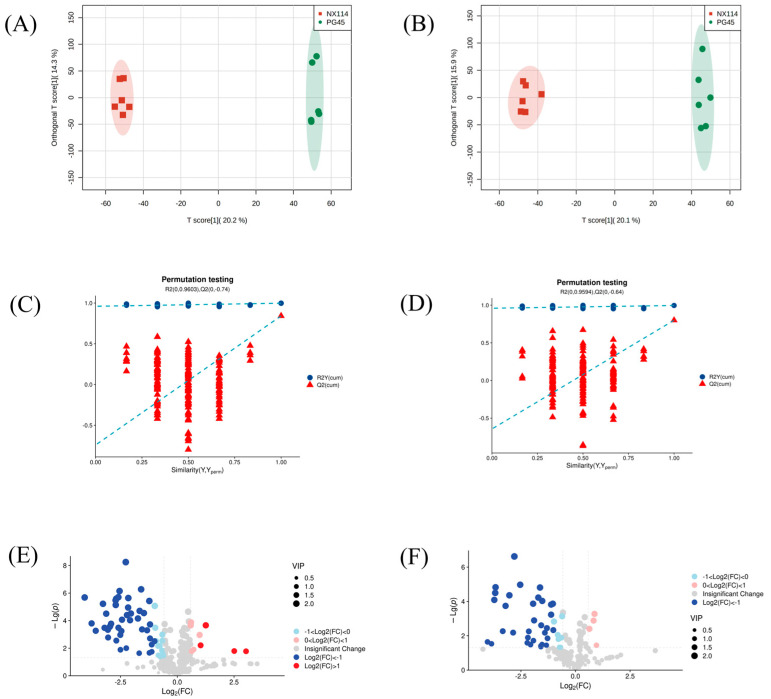
OPLS-DA map, 200-displacement test, and volcano map of *M. bovis* metabolomics analysis (**A**,**C**,**E**) ESI+ model, (**B**,**D**,**F**) ESI− model.

**Figure 5 microorganisms-11-02602-f005:**
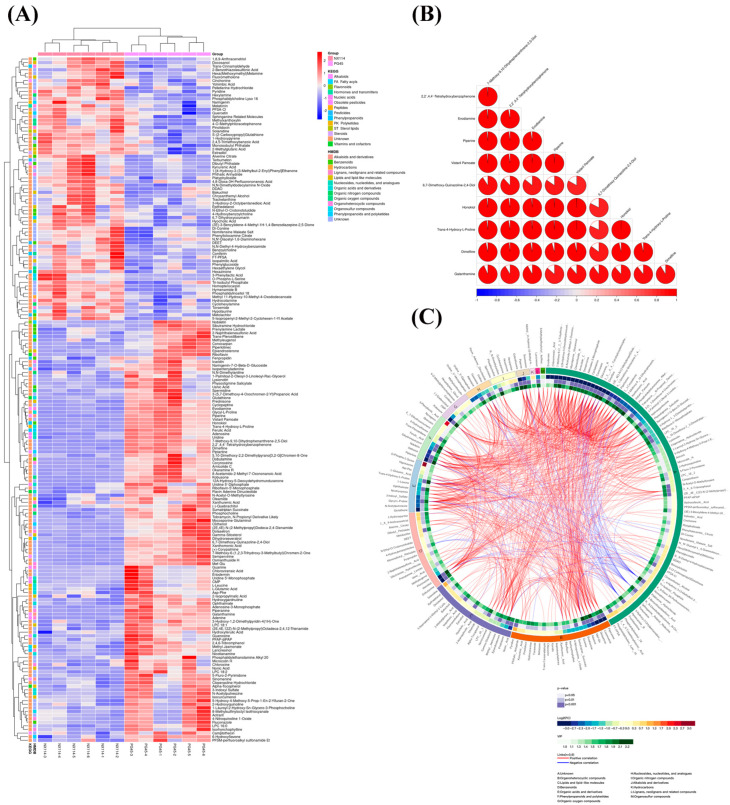
Analysis of differential metabolite expression in *M. bovis* metabolomics. (**A**) Differential metabolite hierarchical clustering, with red indicating greater relative expression levels and blue indicating fewer relative expression levels. The figure shows the metabolite classification information based on the KEGG and HMDB databases. (**B**) The correlation coefficient (**R**) among distinct metabolites lies between −1 and +1 in the correlation matrix heat map. R > 0 denotes a positive correlation, which is depicted in red; R = 0 denotes a negative correlation, which is shown in blue. The bigger the proportion of coloring interval, the stronger the positive or negative correlation. (**C**) Cicos circle, from the outside to the inside, shows the metabolite names, HMDB classification of metabolites, log_2_(fold change), *p* value, VIP, and correlation connections based on OPLS-DA analysis.

**Figure 6 microorganisms-11-02602-f006:**
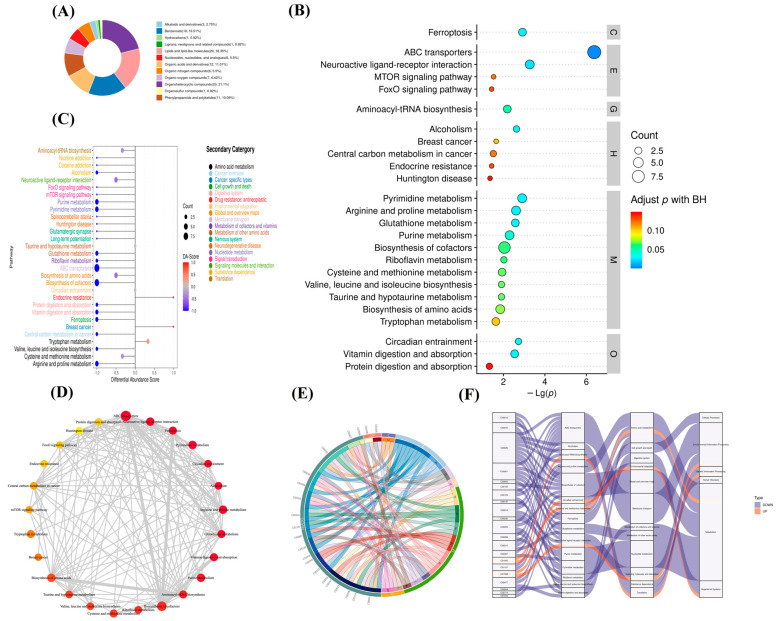
Metabolite function analysis of *M. bovis* metabolomics differences. (**A**). differential metabolite classification ring map (HMDB); (**B**). KEGG pathway enrichment bubble diagram. Level 1 Path classification: Cellular Processes (C), Drug Development (D), Environmental Information Processing (E), Genetic Information Processing (G), Human Diseases (H), Metabolism (M), Organismal Systems (O). (**C**). DA score showed an overall down-regulation trend of metabolic pathways. Functional interaction network diagram of (**D**) channel. (**E**). String diagram shows the correspondence between upregulated metabolites and the KEGG pathway. (**F**). Sankey diagram shows the trend analysis of data flow at the upper down-regulated metabolite and the KEGG pathway levels.

**Figure 7 microorganisms-11-02602-f007:**
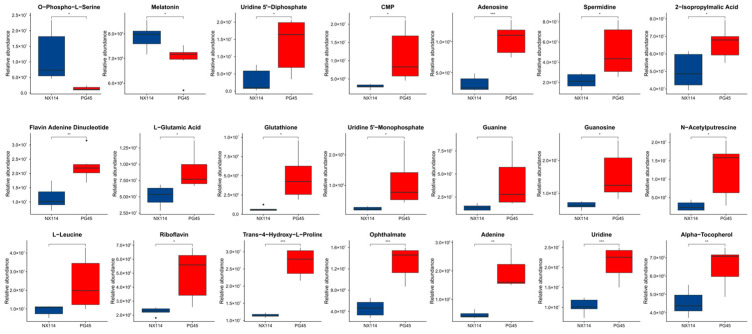
Metabolomic differential metabolites. Welch’s *t*-test was performed for significance, and the specific *p* value was highlighted in the figure and represented by * for *p* < 0.05, ** for *p* < 0.01, *** for *p* < 0.001, and blank for *p* > 0.05 (no significance). The average degree of differential metabolite expression between both groups is also shown, with the horizontal coordinate representing different groups and the vertical coordinate representing the relative expression amount.

## Data Availability

Metabolomics data have been deposited into the EMBL-EBI MetaboLights database with the identifier MTBLS8708.

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
