# Peer review of "UHPLC/MS-Based Untargeted Metabolomics Reveals Metabolic Characteristics of Clinical Strain of Mycoplasma bovis"

_microorganisms, 2023, doi:10.3390/microorganisms11102602_

Round 1
Reviewer 1 Report
This article focuses on untargeted metabolomics studies that reveal the metabolic characteristics of clinical isolates of Mycoplasma bovis. In my opinion, the article is well written, the methods used in the work are modern, and the conclusions correspond to the results obtained. The article also undoubtedly has practical significance. Separately, the quality of the presentation should be noted: the drawings for the article are very well done. I recommend accepting the article in its present form.
Author Response
RE: microorganisms-2572227, research article
Microorganisms
Dear Reviewer,
On behalf of all co-authors, I would like to take this opportunity to thank you and the two reviewers for constructive comments and suggestions on our manuscript (microorganisms-2572227). The manuscript has now been revised according to the three reviewers′ comments and suggestions. In the following, we detail our point-by-point responses to your and the reviewers′ comments and suggestions.
We sincerely hope that the manuscript has been revised to your satisfaction.
Best wishes
Sincerely yours,
Fei Yang, Ph. D.
College of Animal Science and Technology, Ningxia University
E-mail: yangfeiweiwuxian@126.com (Fei Yang)
For review article
Response to Reviewer 1
Open Review
( ) I would not like to sign my review report
(x) I would like to sign my review report
Quality of English Language
(x) I am not qualified to assess the quality of English in this paper
( ) English very difficult to understand/incomprehensible
( ) Extensive editing of English language required
( ) Moderate editing of English language required
( ) Minor editing of English language required
( ) English language fine. No issues detected
Yes Can be improved Must be improved Not applicable
Does the introduction provide sufficient background and include all relevant references?
(x) ( ) ( ) ( )
Are all the cited references relevant to the research?
(x) ( ) ( ) ( )
Is the research design appropriate?
(x) ( ) ( ) ( )
Are the methods adequately described?
(x) ( ) ( ) ( )
Are the results clearly presented?
(x) ( ) ( ) ( )
Are the conclusions supported by the results?
(x) ( ) ( ) ( )
Comments and Suggestions for Authors
General Comments: This article focuses on untargeted metabolomics studies that reveal the metabolic characteristics of clinical isolates of Mycoplasma bovis. In my opinion, the article is well written, the methods used in the work are modern, and the conclusions correspond to the results obtained. The article also undoubtedly has practical significance. Separately, the quality of the presentation should be noted: the drawings for the article are very well done. I recommend accepting the article in its present form.
Response: Thank you very much for taking the time to review this manuscript. We are very grateful to the reviewer for his/her favorable comments on our paper. We also thank you for your suggestion to accept the article. Secondly, responses were also made to the comments of the other two reviewers, and all changes were marked in red and blue font in the revised manuscript.

Reviewer 2 Report

There are numerous linguistic and stylistic errors which often mislead the reviewer.
Author Response
RE: microorganisms-2572227, research article
Microorganisms
Dear Reviewer,
On behalf of all co-authors, I would like to take this opportunity to thank you and the two reviewers for constructive comments and suggestions on our manuscript (microorganisms-2572227). The manuscript has now been revised according to the three reviewers′ comments and suggestions. In the following, we detail our point-by-point responses to the reviewer_2 comments and suggestions.
We sincerely hope that the manuscript has been revised to your satisfaction.
Best wishes
Sincerely yours,
Fei Yang, Ph. D.
College of Animal Science and Technology, Ningxia University
E-mail: yangfeiweiwuxian@126.com (Fei Yang)
For review article
Response to Reviewer 2
Open Review
(x) I would not like to sign my review report
( ) I would like to sign my review report
Quality of English Language
( ) I am not qualified to assess the quality of English in this paper
( ) English very difficult to understand/incomprehensible
(x)Extensive editing of English language required
( ) Moderate editing of English language required
( ) Minor editing of English language required
( ) English language fine. No issues detected
Yes Can be improved Must be improved Not applicable
Does the introduction provide sufficient background and include all relevant references?
(x) ( ) ( ) ( )
Are all the cited references relevant to the research?
(x) ( ) ( ) ( )
Is the research design appropriate?
( ) ( ) (x) ( )
Are the methods adequately described?
( ) ( ) (x) ( )
Are the results clearly presented?
( ) ( ) (x) ( )
Are the conclusions supported by the results?
( ) ( ) (x) ( )
Comments and Suggestions for Authors
General Comments: The manuscript submitted for review presents interesting data on the metabolic characteristics of a field isolate of Mycoplasma bovis. The methods used seem to be justified for the intended purpose, but numerous linguistic and stylistic errors often mislead the reviewer. In the title of the manuscript and in many other places in the manuscript there is variable information about the number of tested isolates, sometimes it mentions one so-called clinical isolate (NX114) and one so-called standard isolate (PG45), and elsewhere the plural is used to refer to the number of so-called clinical isolates, e.g. in the title of the manuscript. Other examples for this case are: lines 12, 72, 77, 294, 365, 392, 451. As for other important remarks, the manuscript is difficult to understand in places, e.g. in many places in the Material and Methods section, it is written in the form of standard instructions/procedures for performing certain tests (e.g. lines 72-76 ‘Inoculate the seed bacterial solution …; 89-94 ‘Inoculate the isolated strain …; ‘Plot the M. bovis growth curve using …’). The whole manuscript requires general linguistic correction.
Response: Thank you very much for taking the time to review this manuscript. We are very grateful to the reviewer for his/her favorable comments on the data of the metabolomics of our manuscript.
Secondly, We are grateful to the reviewer for his/her constructive comments. We revised many sentences that incorrectly described the method. The main changes are as follows:
“2.1 Bacterial strains and media(Lines 64-66). 2.2 Transmission EM(Lines 75-81).
2.3 Determination of growth curve(Lines82-87)2.4 Extraction of metabolites from strains(Lines 88-94)”.
We revised the title of the manuscript and many other places for miswriting regarding the number of isolates tested(e.g. lines 3, 13, 71, 74, 89, 158, 166, 168, 170, 175, 178, 222, 223, 242, 253, 277, 278, 333, 334, 391, 395).
We modified the description of the determination of growth curve of M. bovis. The main changes are as follows:
“We inoculated the isolates of M. bovis NX114 and M. bovis strain PG45 iwere added into PPLO liquid medium (10% w/v). Then the number of colony-forming units (CFU) were measured once per 0 h up to 72 h (n = 6). Finally, we plotted the M. bovis growth curve using the log CFU/mL parameter as an ordinate and the culture time of the bacterial solution as the abscissa”(Lines82-87).
We invited our native English editor to go over the manuscript and correct grammar and spelling errors. We submitted Certificate of editing (PDF:Certificate_of_editing-OKAUV_4_2_oeqmts8pi2).
We thank the referee comments, and responded to the reviewer′s comments point-to-point, as summarized below. All of the changes are marked in red font in the revised manuscript.
Other comments:
Abstract:
Comments: Lines 26-28: this is a conclusion that does not result from this study.
Response: Thanks for your helpful comments. I've changed how I describe this conclusion. The main changes are as follows:
“This study reflects the species-specific differences between two strains of M. bovis and further en-riches our understanding of its metabolism, paving the way for further research into its patho-genesis”(Lines 23-25).
Materials and Methods
Comments: Lines 76-77 the authors should explain how many animals one NX114 isolate was isolated from; it is unclear and misleading, because one isolate cannot come from several calves, but here is such information.
Response: We thank the referee for constructive comments. We have made a more detailed description in accordance with the reviewer′s request. The main changes are as follows:
“M. bovis clinical strain NX114 (GenBank accession no. CP135997) was isolated from the lung of a diseased Holstein calf with pneumonia in Ningxia Province, China. M. bovis type strain PG45 (ATCC 25523) is the international reference strain of M. bovis, and it was donated by Professor Aizhen Guo of Huazhong Agricultural University of China”(Lines 69-74).
Comments: Line 142: the authors should explain what does ‘the two-tailed’ mean in this context
Response: We thank the referee for this good question. “The two-tailed” test is a statistical test used in inference, in which a given statistical hypothesis will be rejected when the value of the statistic is either sufficiently small or sufficiently large. The test is named after the "tail" of data under the far left and far right of a bell-shaped normal data distribution, or bell curve. However, the terminology is extended to tests relating to distributions other than normal.
Results
Comments: Lines 163-165: this part of text should be moved to Materials and Methods section
Response: Thank you for pointing this out. I/We agree with this comment. This sentence has been described in Materials and Methods. Therefore, we delete this sentence.
Comments: Line 190 (for example): the authors should unify in the whole text of manuscript whether they use the division into isolates or groups, which makes it very difficult to interpret the results and read freely.
Response: A unified description of the division of isolates and groups was made.
Comments: Line 250: please correct ,To systemically study …’.
Response: We have corrected this mistake(Line 239).
Comments: Lines 264-272: this sentence is illegible and needs to be thoroughly rebuilt. Most of the information provided here should be moved to the Materials and Methods section.
Response: Thanks for your helpful comments. I have moved the description of the Lines 264-266 method in the original paragraph to the comments in Figure 7(Lines 262-264).
Secondly, We have rewritten this sentence as follows:
“Differential metabolites were observed between the two strains, with upregulated metabolites including O-Phospho-L-Serine (SEP) and melatonin (ML) (p < 0.05), and downregulated metabolites including uridine 5'-diphosphate (UDP), L-glutamic acid (Glu), glutathione (GSH), CMP, uridine 5'-monophosphate (UMP), riboflavin (B2), spermidine (SP), guanosine (Gsn), 2-isopropylmalic acid (2-IPMA) (p < 0.05). Additionally, N-acetylputrescine (NAP), adenine (Ade), alpha-tocopherol (at), flavin adenine dinucleotide (FAD) (p < 0.01), adenosine (Adn), uridine (Urd), trans-4-hydroxy-L-proline (T-4-Hyp), ophthalmate (OA) (p < 0.001), L-leucine (Leu), and guanine (G) did not show significant differences (p > 0.05)”(Line 253-261).
Discussion
Comments: Lines 286-287: please remove 'bovine'; what does mean 'bovine M. bovis'?
Response: Thanks to the reviewer's carefulness, we have corrected this mistake.
Comments: Lines 354-361: please unify the capitalization of the given metabolites.
Response: We have corrected the capitalization of the given metabolites(Lines 313-314).
Comments: Line 373: please write the genus in capital letters and the whole name in italics.
Response: Thanks to the reviewer's carefulness, This sentence was removed in a later revision.
Comments: Line 385: please change 'bovine' to 'bovis' and write the whole name in italics.
Response: Thanks to the reviewer's carefulness, we have corrected this mistake(Line 340).
Comments: Lines 403-405: please correct the sentence.
Response:We have corrected the sentence.
Comments: Line 425: please change ‘outside world’.
Response: I have changed the description of 'outside world' and revised the whole sentence(Lines 375-377).
Comments: In many places in the text, the names of microorganisms should be written in italics (e.g. lines 157, 159, 304, 317, 325, 462-464).
Response: Thanks to the reviewer's carefulness, we have corrected these mistakes(Line 149,151,284,286,340,407,408,409)
Comments: In many places in the manuscript there is no space before the reference bracket or other brackets or places in the text (e.g. lines 38, 41, 44, 46, 104, 119, 128, 280, 285, 288, 383, 387).
Response: Thanks to the reviewer's carefulness, we have made a uniform change.
Comments on the Quality of English Language
There are numerous linguistic and stylistic errors which often mislead the reviewer.
Response: We appreciate your helpful comments. We invited our native English editor to go over the manuscript and correct grammar and spelling errors. Secondly, we submitted Certificate of editing(Certificate_of_editing-OKAUV_4_2_oeqmts8pi2).

Reviewer 3 Report
The subject of this MS is innovative and interesting, where the study is focused on the differences in metabolites and metabolic pathways between a standard strain and a clinical isolate of M.bovis. The experiments are well designed and performed. It appears that the authors intended to differentiate the two strains based on their metabolites, but this was not clearly stated in the original text of the manuscript. Also, the application of the findings of this study should be clearly stated.
The title should be revised. For example, it should be written as “UPLC/MS-based untargeted metabolomics reveals metabolic characteristics of clinical isolates of Mycoplasma bovis”
The abstract is informative.
The aim and hypothesis of the study must be clearly stated in the introduction of the article.
The methods used to perform the study are clear and adequate.
The results presented are adequate.
The figures used to show the results are adequate.
Paragraphs, especially in the discussion sections of the article, often do not have a logical and consistent relationship with each other and cannot well present the intended objectives and results of this study to the reader. In addition, the authors in this section have provided very extensive and unnecessary explanations regarding the parameters measured in this study. Therefore, these sentences should be removed, and the remaining content should be logically connected in a specific process so that it is easy for readers to understand. Moreover, the authors are supposed to explore the reasons and enlarge the discussion in terms of possible mechanisms involved as changes were observed.
The conclusion was supported by the results and clearly expressed the main hypothesis of the study. However, it is suggested that authors make suggestions for future studies.
Author Response
RE: microorganisms-2572227, research article
Microorganisms
Dear Reviewer,
On behalf of all co-authors, I would like to take this opportunity to thank you and the two reviewers for constructive comments and suggestions on our manuscript (microorganisms-2572227). The manuscript has now been revised according to the three reviewers′ comments and suggestions. In the following, we detail our point-by-point responses to the reviewer_3 comments and suggestions.
We sincerely hope that the manuscript has been revised to your satisfaction.
Best wishes
Sincerely yours,
Fei Yang, Ph. D.
College of Animal Science and Technology, Ningxia University
E-mail: yangfeiweiwuxian@126.com (Fei Yang)
For review article
Response to Reviewer 3
Open Review
(x) I would not like to sign my review report
( ) I would like to sign my review report
Quality of English Language
(x) I am not qualified to assess the quality of English in this paper
( ) English very difficult to understand/incomprehensible
( ) Extensive editing of English language required
( ) Moderate editing of English language required
( ) Minor editing of English language required
( ) English language fine. No issues detected
Yes Can be improved Must be improved Not applicable
Does the introduction provide sufficient background and include all relevant references?
( ) (x) ( ) ( )
Are all the cited references relevant to the research?
( ) (x) ( ) ( )
Is the research design appropriate?
(x) ( ) ( ) ( )
Are the methods adequately described?
(x) ( ) ( ) ( )
Are the results clearly presented?
(x) ( ) ( ) ( )
Are the conclusions supported by the results?
(x) ( ) ( ) ( )
Comments and Suggestions for Authors
General Comments: The subject of this MS is innovative and interesting, where the study is focused on the differences in metabolites and metabolic pathways between a standard strain and a clinical isolate of M.bovis. The experiments are well designed and performed. It appears that the authors intended to differentiate the two strains based on their metabolites, but this was not clearly stated in the original text of the manuscript. Also, the application of the findings of this study should be clearly stated.
Response: Thank you very much for taking the time to review this manuscript. We are very grateful to the reviewer for his/her favorable comments on our paper.
Secondly, We conjectured O-Phospho-L-serine (SEP) as a potential signature metabolite and indicator of patho-genicity. SEP has potential as a biomarker with high expression in M. bovis. Whether SEP can be used to distinguish between Mycoplasma of different species,and M. bovis of different genotypes. Whether the accumulation of SEP can be used as an indicator to judge the severity of the disease after M. bovis infection should be further evaluated by metabolite quantitative test to improve understanding of the pathogenesis of M. bovis(Lines 340-345). Future studies will be conducted to further verify this hypothesis.
In addition, the application of the results of the study.The main changes are as follows:
“We believe that this paper will contribute to the understanding of M. bovis biology and its implications for bovine health. Moreover, our findings will stimulate further research in this area and pave the way for novel therapeutic strategies”(Lines 64-66).
We thank the referee for favorable comments, and responded to the reviewer′s comments point-to-point, as summarized below. All of the changes are marked in blue font in the revised manuscript.
Specific Comments:
Comments1: The title should be revised. For example, it should be written as “UPLC/MS-based untargeted metabolomics reveals metabolic characteristics of clinical isolates of Mycoplasma bovis”
Response1: We appreciate your helpful comments. According to the referee′s suggestion, we have revised the title of the article. The main changes are as follows:
“UHPLC/MS-based untargeted metabolomics reveals metabolic characteristics of clinical strain of Mycoplasma bovis”
Comments2: The abstract is informative.
Response2: We are very grateful to the reviewer for his/her favorable comments. We supplemented the impact and problem statement at the beginning of the abstract. The sentences added are as follows:
“Mycoplasma bovis is a global concern for the cattle industry owing to its high rates of infection and resulting morbidity, but its pathogenesis remains poorly understood”(Lines 11-12).
Additionally, we reduced the word count to comply with the 200-word maximum specified by the journal(Lines 11-25).
Comments3: The aim and hypothesis of the study must be clearly stated in the introduction of the article.
Response3: We thank the referee for constructive comments. We supplemented the study objectives and hypotheses in the introduction. The sentences added are as follows:
“We believe that this paper will contribute to the understanding of M. bovis biology and its implications for bovine health. Moreover, our findings will stimulate further research in this area and pave the way for novel therapeutic strategies”(Lines 64-66).
Comments4: The methods used to perform the study are clear and adequate.
Response4: We thank the referee for favorable comments.
Comments5: The results presented are adequate.
Response5: We are very grateful to the reviewer for his/her favorable comments on the description of the results of our paper.
Comments6: The figures used to show the results are adequate.
Response6: We are very grateful to the reviewer for his/her favorable comments.
Comments7: Paragraphs, especially in the discussion sections of the article, often do not have a logical and consistent relationship with each other and cannot well present the intended objectives and results of this study to the reader. In addition, the authors in this section have provided very extensive and unnecessary explanations regarding the parameters measured in this study. Therefore, these sentences should be removed, and the remaining content should be logically connected in a specific process so that it is easy for readers to understand. Moreover, the authors are supposed to explore the reasons and enlarge the discussion in terms of possible mechanisms involved as changes were observed.
Response7: We are grateful to the reviewer for his/her constructive comments.
According to the reviewer's suggestion, first of all, we rearranged the discussion section of the revised manuscript and added subheadings to better present the expected objectives and results of this study to readers. The main changes are as follows:
Part 1:A discussion of the significance of the obtained results and why they are relevant in the present study in light of the objectives.
Our new subheadings are:“4.1 Important components of metabolites of M. bovis clinical strain”(Line 284). “4.2 Trend of differential metabolites and pathways at all levels of M. bovis”(Line 311). “4.3 Potential signature metabolite and possible mechanisms of M. bovis”(Line 326). “4.4 Nucleic acid metabolite differences reflect distinct growth characteristics of M. bovis”(Line 359). “4.5 Significant involvement of the transport system in M. bovis metabolism”(Line 377).
Second, we removed the very extensive and unnecessary explanations in the discussion, streamlining the entire discussion section to make it easy for readers to understand the causes of differential metabolites and the possible mechanisms involved, further enriches our understanding of its metabolism, paving the way for further research into its pathogenesis(Lines 271-396).
Comments8: The conclusion was supported by the results and clearly expressed the main hypothesis of the study. However, it is suggested that authors make suggestions for future studies.
Response8: We are very grateful to the reviewer for his/her favorable comments on the description of the conclusion of our paper. We make recommendations for future research. The sentences added are as follows:
“These findings will stimulate further research in this area and pave the way for novel therapeutic strategies”(Lines 406-407).

Round 2
Reviewer 2 Report
The authors addressed all the reviewer's comments. I have three minor comments as follows:
Line 83: please correct 'iwere'
Line 284: please change Mycoplasma pmeumonia to Mycoplasma pneumoniae; 'ac-curately' to 'accurately'.
Reviewer 3 Report
-